# SciGram: A Large-Scale Dataset for Scientific Diagram Understanding

## Abstract

Multimodal large language models (MLLMs) have achieved strong results in visual question answering with natural images, yet their performance on diagram-based reasoning remains limited, largely due to the scarcity of high-quality training data. We present **SciGram**, a dataset of 200,000 scientific diagrams paired with synthetic instruction-following data grounded in middle-school science terminology. SciGram is built through a cost-efficient pipeline for generating multimodal alignment and instruction data tailored to vision-language understanding with scientific diagrams. Fine-tuning LLaVA on SciGram (**LLaVA-SciGram**) consistently improves diagram-based question answering on TQA, ScienceQA, and AI2D, surpassing LLaVA-OneVision with substantially fewer instructions. Furthermore, incorporating SciGram as an additional instruction-tuning stage for LLaVA-OneVision establishes new state-of-the-art results across all three benchmarks, underscoring the robustness and effectiveness of our dataset. To foster progress in diagram understanding, we release both the SciGram dataset and the LLaVA-SciGram model.

## 1 Introduction

In his 1988 AAAI Presidential Address, Raj Reddy highlighted a fundamental AI Grand Challenge: answering textbook-style questions that integrate vision, language, reasoning, and learning (Reddy, 1988). Today this challenge remains open, particularly in natural sciences, where concepts such as photosynthesis, the water cycle, or energy transfer are presented through both text and diagrams.

Benchmarks like Textbook Question Answering (TQA) (Kembhavi et al., 2017) address this challenge by posing multimodal questions that require reasoning over both textual passages and scientific diagrams. However, despite progress in visual language models (VLMs), diagram understanding is still a difficult task. Scientific diagrams are symbolic, abstract, and structurally diverse, designed to convey functional or relational meaning rather than depict literal scenes. Unlike natural images, which modern VLMs are trained on at scale, diagrams demand spatial reasoning, interpretation of arrows, labels, and domain-specific conventions.

A major bottleneck is the lack of large-scale, high-quality training data tailored to scientific diagrams. Existing visual instruction datasets largely focus on natural or web-based imagery, leaving a gap in domain-specific, multimodal supervision. To address this, we introduce SciGram, a large-scale dataset of over 200,000 scientific diagrams paired with synthetic instruction-following data. Grounded in terminology from middle-school science textbooks, SciGram enables the generation of semantically rich vision-language pairs for fine-tuning VLMs for scientific diagram understanding.

Our contributions are as follows:

- **A scalable methodology** for collecting multimodal data using science-specific terminology to retrieve diagrams from the Web.

- **The SciGram dataset**: a language-vision dataset of diagrams and synthetically generated instructions in the domain of natural sciences.

- **The LLaVA-SciGram models**: a family of VLMs fine-tuned on SciGram, based on the LLaVA architecture (Liu et al., 2023; 2024; Xu et al., 2025; Li et al., 2025).

- **A comprehensive evaluation** on diagram-centric QA benchmarks, outperforming or on par with state-of-the-art VLMs.

The rest of the paper is organized as follows: Section 2 reviews related work. Section 3 outlines our data collection methodology. Section 4 describes the SciGram dataset in detail. Section 5 presents our models and evaluation results. Section 6 concludes with future directions. The code used to generate the dataset, train the models and evaluate them is publicly available[1].

## 2 BACKGROUND

Early work on diagram understanding such as TQA (Kembhavi et al., 2017) explored approaches ranging from machine comprehension models (BiDAF (Seo et al., 2017), MemoryNet (Weston et al., 2014)) to early VQA methods (Antol et al., 2015) and diagram-specific parsers (DSDP-NET (Kembhavi et al., 2016b)). These efforts established that diagrams pose challenges distinct from natural images, with generalization across heterogeneous diagram types proving particularly difficult.

Subsequent advances took two directions. Reasoning-focused approaches, e.g., IGMN (Li et al., 2018), emphasized contradiction detection, while graph-based models such as f-GCN (Kim et al., 2019) and RAFR (Ma et al., 2021), captured spatial and semantic relations. In parallel, transformer-based methods improved text-based QA, including ELMo (Peters et al., 2018), BERT (Devlin et al., 2018), RoBERTa (Liu et al., 2019a), and PaLM (Chowdhery et al., 2022), and extended to multimodal QA, such as VL-BERT (Su et al., 2019), and LXMERT (Tan & Bansal, 2019). However, pretrained on broad image–text corpora, such as Conceptual Captions (Sharma et al., 2018), MS COCO (Lin et al., 2014), and Visual Genome (Krishna et al., 2017), these models lacked the structured semantics required for scientific diagrams. ISAAQ (Gomez-Perez & Ortega, 2020) addressed this gap via cross-attention between modalities, achieving significant but still limited gains.

Contrastive approaches like CLIP (Radford et al., 2021) and SIGLIP (Zhai et al., 2023) further advanced pretraining by jointly aligning image–text embeddings at scale, forming the backbone of modern multimodal architectures. Building on this, multimodal LLMs such as LLaVA (Liu et al., 2023), and Molmo (Deitke et al., 2024) integrate visual encoders with LLMs and fine-tune on synthetic instruction-following data. Their training relies on large-scale general-purpose datasets like LLaVA-OneVision (Li et al., 2025) and PixMo (Deitke et al., 2024)), with only limited coverage of diagram-centric benchmarks such as TQA (Kembhavi et al., 2017), AI2D (Kembhavi et al., 2016b), and ScienceQA (Lu et al., 2022b). As a result, progress in scientific diagram QA remains constrained by the lack of large, high-quality, domain-specific data. Domain-specialized MLLMs like LLaVA-Med (Li et al., 2023), LLaVA-Chef (Mohbat & Zaki, 2024), and LLaVA-Ultra (Guo et al., 2024) demonstrate the value of curated datasets, showing that scaling to scientific diagrams is feasible but data-limited.

In this work, we target two persistent gaps: (i) the absence of large-scale, high-quality diagram datasets, and (ii) the scarcity of instruction-following data aligned with text–diagram semantics. To address these challenges, we introduce SciGram, a synthetic multimodal dataset tailored to scientific diagrams, enabling more robust multimodal scientific reasoning beyond existing benchmarks.

## 3 APPROACH

We propose a cost-efficient methodology to build an instruction-following multimodal dataset for diagram understanding. Using a lightweight and flexible architecture like LLaVA (Liu et al., 2023), we adapt it to middle-school science. Aligned with LLaVA's two-stage pipeline (alignment and visual instruction fine-tuning), we introduce a six-step process (Figure 1) to construct three subsets that support training.

First, we extract domain-specific terminology from a knowledge source, using the TQA middle-school science textbook (Kembhavi et al., 2017). Second, we generate scientific claims from an LLM based on this terminology. Third, these claims are used as search queries to retrieve and curate diagrams from the Web. Fourth, we use a multimodal LLM to generate textual descriptions for each

---

[1]https://github.com/anonymous-sciclaims/scigram

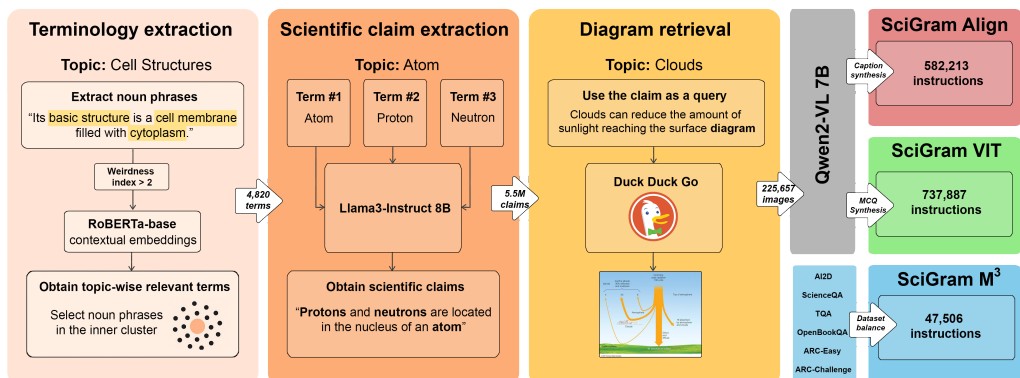

Figure 1: Overview of our six-stage dataset creation pipeline, comprising: terminology extraction, claim generation, diagram retrieval, and synthesis of SciGram subsets (Align, VIT, and M$^3$).

diagram to produce the **SciGram Align** subset, enabling the alignment of visual and textual modalities. Fifth, we synthesize multiple-choice questions grounded in the diagrams to create the **SciGram VIT** subset, enabling model fine-tuning in a question-answering setup. Finally, we compile existing datasets curated by subject matter experts to produce SciGram M$^3$ (MultiModal Mixture), shuffling answer choices to reduce bias. Each step is detailed in the following subsections.

## 3.1 TERMINOLOGY EXTRACTION

To extract a vocabulary representing core scientific concepts, we perform a statistical analysis of the natural science textbook compiled by (Kembhavi et al., 2017) in the TQA dataset. Leveraging the textbook's organization by scientific topics, we analyze each section, including lessons, explanations, and instructional materials, to capture terms associated with distinct concepts. Our method follows three main steps:

1. **Tokenization and Noun Phrase Identification.** We begin by tokenizing the textbook text for each topic and extracting noun phrases, which capture concepts and entities that are likely to be significant in the context of science education.

2. **Selection based on Weirdness Index.** To refine the candidate terms, we compute the weirdness index (Ahmad et al., 1999), comparing term frequencies between a general corpus (BNC[2]) and the domain-specific textbook. Terms exceeding a threshold are retained as distinctive; we empirically set $t = 2$ to capture relevant terms while excluding stop-words. Finally, we lemmatize the filtered noun phrases to unify morphological variants.

3. **Embedding Representation and Clustering.** For each term, we generate contextual embeddings using RoBERTa-base (Liu et al., 2019b) with its surrounding sentence as input. Multiple occurrences within the same topic are averaged. To retain topic-relevant terms, we compute the centroid of all embeddings and discard those whose Euclidean distance exceeds one standard deviation above the mean, yielding a semantically coherent set for each topic.

This multi-stage filtering and clustering process results in a curated vocabulary of **4,820 unique scientific terms** that are both semantically coherent and representative of the middle-school science curriculum of reference. By focusing on high-relevance noun phrases that are statistically and contextually significant, we ensure that the extracted terminology forms a solid foundation for generating accurate and educationally aligned claims in the following stage.

## 3.2 SCIENTIFIC CLAIM GENERATION

With the core terminology extracted, the next step is to generate a diverse set of scientific claims that can serve as queries for retrieving relevant diagrams. To this end, we use the LLaMA3 8B

---

[2]https://www.english-corpora.org/bnc/

Instruct model[3]. We sample the model to synthesize concise, atomic-level (no more than 12 tokens) scientific facts that are appropriate for middle-school learners (e.g., 'protons and neutrons are located in the nucleus of an atom', 'photosynthesis occurs in the chloroplasts of plant cells'). These claims combine the extracted terms to cover a wide range of factual relationships in life, earth, and physical sciences. A prompt template for scientific claim generation is available in A.1.

For each topic from the TQA textbook, the model is prompted to generate multiple batches of 50 claims based on different combinations of the extracted relevant terms. For example, if a topic includes terms {'atom', 'proton', 'neutron'}, we prompt the model to generate 50 claims for each subset—single terms ({'atom'}, {'proton'}, {'neutron'}) as well as pairwise and triplet combinations ({'atom', 'proton'}, {'atom', 'neutron'}, {'proton', 'neutron'}, {'atom', 'proton', 'neutron'})—ensuring broad conceptual coverage. All generated claims are constrained to be concise, meaningful, grounded in common sense reasoning, and suitable for middle-school comprehension.

After generating claims across all topics, we apply a post-processing step to eliminate duplicates and semantically redundant entries. The final output is a collection of **5,508,218 unique scientific claims**, which serve as the textual backbone for retrieving relevant visual content in the next stage.

## 3.3 DIAGRAM RETRIEVAL

To link textual claims with corresponding visual representations, we use each scientific claim as a query to retrieve diagrams from the web. Specifically, we query the DuckDuckGo search engine[4], collecting the top five resulting images of each claim along with their URLs and metadata. [5]

This approach yields candidate diagrams that visually reflect the scientific content of the claims. To ensure relevance and visual quality, we apply several filtering steps to the retrieved images. First, we retain only images that are linked to at least five distinct claims, increasing the likelihood that they represent relevant scientific content. Second, we remove duplicates using perceptual hashing (National Institute of Standards and Technology, 2012) to eliminate identical images. Third, we discard invalid files, such as corrupted images or unsupported formats, e.g., GIFs. After all filtering and validation steps, we obtain a dataset of **255,657 unique images**, each associated with at least five scientific claims. Examples of queries to DuckDuckGo can be found in Appendix A.3.

## 3.4 CAPTION SYNTHESIS

With a curated set of diagrams in place, the next step is to generate rich, descriptive captions that capture their visual content in order to align textual and visual features. To accomplish this, we use the Qwen2-VL-7B model[6], a vision-language model with open weights that we empirically proved to be capable of producing detailed, context-aware textual descriptions from images. For each diagram, the model is instructed to generate a caption in paragraph form, focusing on key components, their relationships, and relevant spatial, temporal or dynamic aspects. A prompt template for caption generation is available in A.1.

To increase diversity and reduce potential bias , we repeat the captioning process three times for each diagram, generating three distinct captions. These image-caption pairs are then formatted into instruction-following examples using a naïve expansion strategy similar to the one proposed in (Liu et al., 2023). Examples of this expansion can be found in Appendix A.4. The resulting dataset, comprising multimodal pairs that support alignment between visual and textual modalities, forms the first training subset, which we refer to as **SciGram Align**.

## 3.5 MULTIPLE-CHOICE QUESTION SYNTHESIS

To support fine-tuning for downstream tasks such as visual question answering, we generate multiple-choice questions (MCQs) grounded in the diagram content. Using the Qwen2-VL-7B model, we prompt the generation of five distinct MCQs per diagram, each with four answer choices.

---

[3]https://huggingface.co/meta-llama/Meta-Llama-3-8B-Instruct
[4]https://pypi.org/project/duckduckgo-search/
[5]We do not claim ownership on the images collected. Licensing information can be found in Appendix A.2
[6]https://huggingface.co/Qwen/Qwen2-VL-7B

The model is instructed to rely solely on visual elements present in the image, ensuring that the questions are directly answerable from the diagram without requiring external knowledge. The questions span the domains of life, earth, and physical sciences, and are phrased at a comprehension level appropriate for middle-school learners, aligning with the educational scope of our target domain. A prompt template for question generation is available in A.1.

To maintain dataset quality, we ensure a balanced distribution of correct answer letters across the corpus. All questions are converted into an instruction-following format, where the expected output is a structured JSON dictionary mapping the correct answer to its letter, e.g., {"answer": "b"}. This machine-friendly representation facilitates consistent training and evaluation. The resulting dataset pairs diagrams with MCQs and answers, forming the second subset, **SciGram VIT**, which supports the transition from basic alignment (captions) to reasoning-based tasks in scientific diagram understanding.

### 3.6 CURATED DATASETS COLLECTION

Following the LLaVA-OneVision training regime (Li et al., 2023), we introduce a further fine-tuning step using curated domain-specific datasets. To this end, we construct the **SciGram $M^3$** subset, which integrates both diagram-based multiple-choice datasets, ScienceQA (Lu et al., 2022a), AI2D (Kembhavi et al., 2016a), and TQA (Kembhavi et al., 2017), and selected text-only QA datasets, such as ARC (Clark et al., 2018) and OpenBookQA (Mihaylov et al., 2018), to reinforce general reasoning abilities. All questions are transformed into instruction format for consistency with the training setup and their answer choices are shuffled across the datasets to ensure a balanced distribution and avoid biases and overfitting.

## 4 THE SCIGRAM DATASET

### 4.1 TERMINOLOGY

We analyze the scientific terminology used to construct the SciGram dataset. Table 1 presents the most and least frequent terms from our final set of 4,820 extracted terms, based on their occurrence in the TQA textbook. Among these, 1,295 terms appear only once, reflecting a long-tail distribution. This skew is mitigated by the weirdness index filter, which retains contextually important terms even if they are infrequent in the source text. In total, 15.09% of candidate noun phrases were eliminated by this filter. Table 2 shows terms with the highest and lowest weirdness-index scores; terms with infinite weirdness-index (appearing in TQA but absent in the BNC corpus) are not included. Distinctive scientific terms such as "Cellular Respiration" and "Epicenter" score highly, as expected.

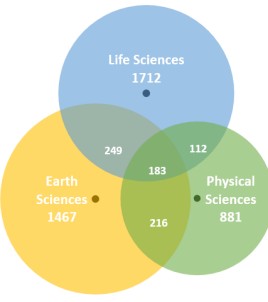

Figure 2: Terminology distribution by subjects. For each subject we include the number of terms that appear in lessons from that specific subject. In the intersections, we report the number of terms belonging to two or three matters at the same time. The total number is 4,820.

| Term | Frequency | Term | Frequency |
|---|---|---|---|
| 1. Water | 1,930 | (other 1291 terms) | 1 |
| 2. Energy | 1,712 | 4817. Gas Increase | 1 |
| 3. Air | 770 | 4818. Temperature and Volume | 1 |
| 4. Earth | 630 | 4819. Total Spread | 1 |
| 5. Body | 586 | 4820. Typical Measurement | 1 |

Table 1: Most and least frequent terms in our terminology selection.

| Term | w-index | Term | w-index |
|---|---|---|---|
| 1. Mechanical Advantage | 20,380.0 | 4816. Invention | 2.03 |
| 2. Cellular Respiration | 18,391.7 | 4817. Shaft | 2.02 |
| 3. Air Mass | 5,799.2 | 4818. Valley | 2.01 |
| 4. Epicenter | 5,560.1 | 4819. Heavy Metal | 2.01 |
| 5. Decomposers | 4,604.5 | 4820. Rubbing | 2.01 |

Table 2: Terms with the highest and lowest weirdness index in our terminology selection. Infinite weirdness index terms were excluded.

Figure 2 shows the subject-wise distribution of terminology across the TQA textbook. While Physical Sciences are slightly underrepresented compared to Earth and Life Sciences, there is substantial

overlap between subjects: 399 terms are shared between Physical and Earth Sciences, and 432 between Earth and Life Sciences. In general, most terms are assigned to a single subject, but nearly 16% appear in two subjects, and 183 terms are shared across all three, highlighting both the specificity and the transversal nature of scientific terminology within the middle-school curricula.

## 4.2 DATASET STRUCTURE

The SciGram dataset consists of three main subsets: SciGram Align, SciGram VIT and SciGram $M^3$. Each subset is designed to be used at each stage of the training pipeline, and therefore has different characteristics we present next.

**SciGram Align** contains **582,213** instructions designed to align visual and text features during the first training stage through a captioning task. Figure 6 shows an example from this subset. The captions consist of one paragraph providing a detailed description of the entities and processes involved in the diagram. Each diagram has 3 captions. More examples can be found at Appendix A.6.

**SciGram VIT** consists of **737,887** instructions created to finetune the model for a Multiple-Choice Question Answering downstream task involving diagrams. An example question can be found in Figure 6. These questions always include 4 possible answer options, which were post-processed to be statistically balanced across the dataset. More examples available in Appendix A.6.

**SciGram $M^3$** consists of **47,506** instructions collected from the TQA training set (14,050 questions), AI2D (12,403), using the same split as LLaVA-OneVision[7], the training set of ScienceQA (12,726), the OpenBookQA training set (4,957), and the ARC-Easy and ARC-Challenge training sets (3,370).

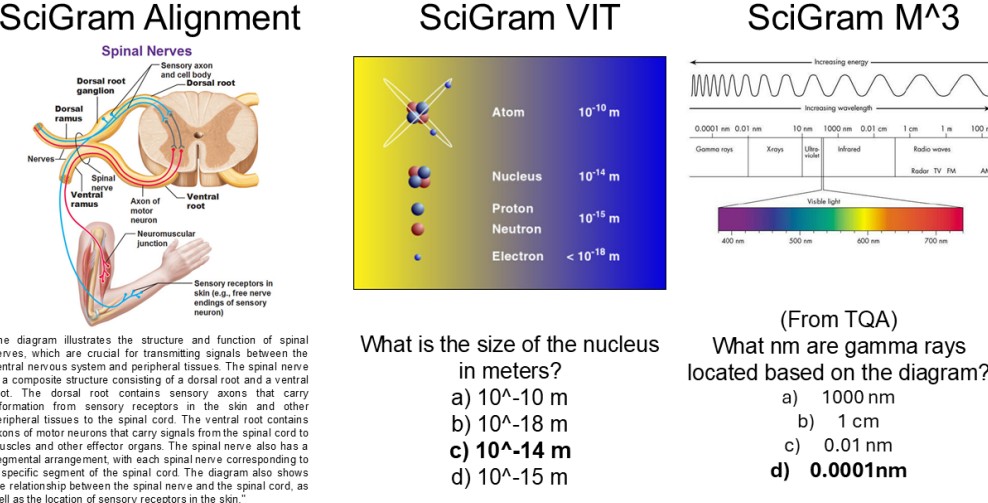

Figure 3: Examples of SciGram subsets.

Figure 4 shows the size comparison between the SciGram dataset and OneVision single-image standard training data across all the LLaVA pipeline. Compared to OneVision data (7.8M instructions), SciGram's focus on scientific diagram understanding allows for a considerably more compact data volume (1.4M instructions). This is crucial to ensure domain relevance and optimize time and computational resources to finetune a MLLM in domain-specific scenarios.

## 5 EXPERIMENTATION

We evaluate the impact of SciGram by fine-tuning a LLaVA model on three diagram-based question benchmarks: TQA, ScienceQA, and AI2D. To assess the model's multimodal capabilities, we also include results on text-based questions in TQA and ScienceQA. To better understand the contribu-

---

[7]https://huggingface.co/datasets/lmms-lab/LLaVA-OneVision-Data

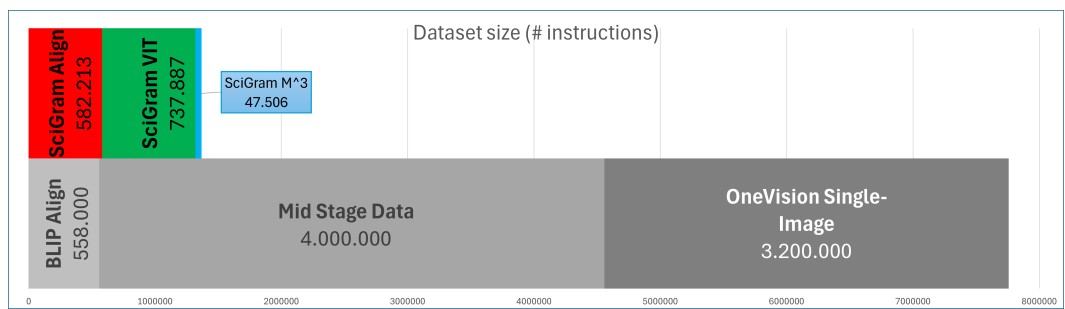

Figure 4: Comparison between SciGram and OneVision training data (number of instructions).

tion of our dataset, we perform an ablation study analyzing the effects of SciGram and its individual subsets across all benchmarks.

## 5.1 MODELS AND TRAINING

We follow two training regimes: (i) using only SciGram subsets in each training phase and (ii) using SciGram VIT and SciGram $M^3$ to finetune a pre-trained LLaVA OneVision 7B model. We name the resulting models as LLaVA-SciGram 7B and LLaVA-SciGram OneVision 7B, respectively.

The architecture of **LLaVA-SciGram 7B** is formed by a pretrained CLIP vision branch and a Qwen2 Instruct 7B as the LLM. For training, we follow the same pipeline as LLaVA OneVision. First, an alignment stage where the image features are aligned with the pre-trained LLM word embedding. We train the projection matrix using the instructions from SciGram Align, keeping both the visual encoder and LLM weights frozen, during one epoch and with a learning rate of 1e-3. Second, a instruction tuning stage using LoRA to train on the SciGram VIT subset during one epoch and learning rate 1e-5. After merging the LoRA adapter into the model, we use SciGram $M^3$ to fine-tune another LoRA adapter during 3 epochs and with a learning rate of 1e-5.

The **LLaVA-SciGram OneVision 7B** model uses the architecture and weights from LLaVA OneVision Single Image training, formed by SIGLIP as the vision branch, and Qwen2 Instruct 7B as the LLM. We train on SciGram VIT and SciGram $M^3$ as mentioned above, skipping the alignment step.

All training was done on two NVIDIA A100 GPUs. Hyperparameters can be found in Appendix A.5

## 5.2 RESULTS

### 5.2.1 TQA

The TQA dataset consists of text-only multiple-choice and true/false questions, as well as diagram-grounded questions, about middle-school science curricula. We evaluate our models against different baselines: (i) TQA literature baselines, a set of state-of-the-art models from the literature, (ii) large unified multimodal models with similar sizes, such as Phi-3 Vision(Abdin et al., 2024), Molmo 7BDeitke et al. (2024), Pixtral 12B(Agrawal et al., 2024) or Qwen2-VL 7B(Yang et al., 2024), (iii) Multimodal APIs, such as Gemini 1.5 Flash(Team et al., 2024) and GPT4o(OpenAI, 2024), and (iv) other LLaVA models. Since some of these models have been trained on not publicly available data that may include the TQA test set, we shuffled the answer options for each question.

As shown in Table 3, our model based on a pretrained LLaVA OneVision model and finetuned with SciGram VIT and SciGram $M^3$ obtains the current state-of-the-art in diagram-based questions, outperforming all the baselines. Moreover, both LLaVA-SciGram and LLaVA-SciGram OneVision improve over the general and diagram-based results of LLaVA OneVision. The best general model for this benchmark is GPT4o, thanks to its good results in text-based questions.

### 5.2.2 SCIENCEQA

ScienceQA is collected from elementary and high school science curricula, and contains multimodal multiple-choice science questions that can include diagram questions, text-only questions, and also

| Model | Text MC | Text TF | Diagram MC | All |
|---|---|---|---|---|
| Random | 22.88 | 50.10 | 24.96 | 29.08 |
| MemN+VQA | 31.05 | 50.50 | 31.82 | 35.11 |
| MemN+DPG | 30.98 | 50.50 | 32.83 | 35.62 |
| BiDAF+DPG | 30.46 | 50.40 | 32.72 | 35.39 |
| FCC+Vecsigrafo | 36.56 | - | 35.30 | - |
| IGMN | 40.00 | 57.41 | 36.35 | 41.36 |
| f-GCN1+SSOC | 49.54 | 62.73 | 37.61 | 45.77 |
| ISAAQ | 72.06 | 78.83 | 55.12 | 64.66 |
| Phi-3 Vision* | 81.19 | 74.01 | 72.91 | 75.52 |
| MOLMo 7B-D* | 81.35 | 84.80 | 71.20 | 76.69 |
| Pixtral 12B* | 85.84 | 91.85 | 77.08 | 82.39 |
| Qwen2-VL 7B* | 87.69 | 91.63 | 78.08 | 83.41 |
| Gemini 1.5 Flash* | 90.94 | 94.73 | 68.18 | 79.73 |
| GPT4o* | **94.20** | **96.16** | 77.32 | **85.74** |
| LLaVa 1.5 7B* | 67.03 | 60.02 | 39.85 | 51.47 |
| LLaVA OneVision 7B* | 89.09 | 88.49 | 77.08 | 82.70 |
| LLaVA-SciGram 7B (ours)* | 90.87 | 92.00 | 76.68 | 83.66 |
| LLaVA-SciGram OneVision 7B (ours)* | 90.56 | 92.87 | **80.21** | 85.59 |

Table 3: TQA test set accuracies. Obtained by us with custom prompts: *; rest, from literature.

questions with natural images, offering a wider perspective on the different modalities involved in scientific domain. Some questions also include hints about the question and/or the image.

| Model | NAT | SOC | LAN | TXT | IMG | NO | G1-6 | G7-12 | Avg |
|---|---|---|---|---|---|---|---|---|---|
| Human | 90.23 | 84.97 | 87.48 | 89.60 | 87.50 | 88.10 | 91.59 | 82.42 | 88.40 |
| MCAN (Yu et al., 2019) | 56.08 | 46.23 | 58.09 | 59.43 | 51.17 | 55.40 | 51.65 | 59.72 | 54.54 |
| Top-Down (Anderson et al., 2018) | 59.50 | 54.33 | 61.82 | 62.90 | 54.88 | 59.79 | 57.27 | 62.16 | 59.02 |
| BAN (Kim et al., 2018) | 60.88 | 46.57 | 66.64 | 62.61 | 52.60 | 65.51 | 56.83 | 63.94 | 59.37 |
| DFAF (Gao et al., 2018) | 64.03 | 48.82 | 63.55 | 65.88 | 64.49 | 64.11 | 57.12 | 67.17 | 60.72 |
| ViLT (Kim et al., 2021) | 60.48 | 63.89 | 60.27 | 63.20 | 61.38 | 57.00 | 60.72 | 61.90 | 61.14 |
| Patch-TRM (Lu et al., 2022c) | 65.19 | 46.79 | 65.55 | 66.96 | 55.28 | 64.95 | 58.04 | 67.50 | 61.42 |
| VisualBERT (Li et al., 2019) | 59.33 | 69.18 | 61.18 | 62.71 | 62.17 | 58.54 | 62.96 | 59.92 | 61.87 |
| UnifiedQA Base (Khashabi et al., 2020) | 68.16 | 69.18 | 74.91 | 63.78 | 61.38 | 77.84 | 72.98 | 65.00 | 70.12 |
| GPT-4 w/ CoT (OpenAI, 2024) | 85.48 | 72.44 | 90.27 | 82.65 | 71.49 | 92.89 | 86.66 | 79.04 | 83.99 |
| LLaMA-Adapter (Zhang et al., 2024) | 84.37 | 88.30 | 84.36 | 83.72 | 80.32 | 86.90 | 85.83 | 84.05 | 85.19 |
| Chameleon (Lu et al., 2023) | 89.83 | 74.13 | 89.82 | 88.27 | 77.64 | 92.13 | 88.03 | 83.72 | 86.54 |
| LaVIN-13B (Luo et al., 2023) | 89.88 | 94.49 | 89.92 | 88.95 | 87.61 | 91.85 | 91.45 | 89.72 | 90.83 |
| KAM-CoT (Mondal et al., 2024) | 94.76 | 82.24 | 93.36 | 94.53 | 93.16 | 94.15 | 94.24 | 93.21 | 93.87 |
| T-SciQ (Wang et al., 2024) | **96.89** | 95.16 | **95.55** | 96.53 | 94.70 | **96.79** | **96.44** | 95.72 | **96.18** |
| LLaVA OneVision 7B* | 88.10 | 88.98 | 78.64 | 92.40 | 87.31 | 80.14 | 88.95 | 80.22 | 85.83 |
| LLaVA-SciGram 7B (ours)* | 96.27 | 97.53 | 91.64 | **99.11** | 95.24 | 93.38 | 95.49 | 95.06 | 95.33 |
| LLaVA-SciGram OneVision 7B (ours)* | **97.65** | **98.31** | 90.82 | 98.48 | **97.52** | 92.54 | 96.15 | **95.78** | 96.02 |

Table 4: ScienceQA test accuracies. By us, using custom prompts: *; rest, literature results. NAT: Natural Sciences. SOC: Social. LAN: Language. TXT: Questions with text support. IMG: visual support. NO: No support. G1-6: Grade 1 to Grade 6. G7-12: Grade 7 to Grade 12. AVG: average.

As shown in Table 4, our two models are the new state-of-the-art in ScienceQA for questions with visual support. Furthermore, LLaVA-SciGram OneVision is on par with the current best general result, T-SciQ (Wang et al., 2024), which uses chain-of-thought to refine its results.

### 5.2.3 AI2D

AI2D contains grade school science diagrams with multiple choice questions. Since AI2D does not have an official test set, we used the most used split in the literature[8]. This subset includes diagrams with obscured labels along questions that require the model to deeply understand the elements and processes in the image. There is another, less demanding version of this split based on the same questions, where diagrams have transparent labels[9], which we also include in the evaluation.

Table 5 presents results in both splits. LLaVA-SciGram OneVision model surpasses Molmo, the previous state-of-the-art model in the split with opaque labels, showing the robustness and the ability

---

[8]https://huggingface.co/datasets/lmms-lab/ai2d
[9]https://huggingface.co/datasets/lmms-lab/ai2d-no-mask

| Model | Opaque labels | Transparent labels |
|---|---|---|
| Phi-3 Vision | 74.19* | 78.10 |
| MOLMO 7B-D | 82.40 | 93.20 |
| Pixtral 12B | 76.46* | 79.00 |
| Qwen2-VL 7B | 80.57* | 83.00 |
| Gemini 1.5 Flash | 73.01* | 81.30* |
| GPT4o | 74.61* | **94.20** |
| LLaVA OneVision 7B | 79.50* | 89.90 |
| LLaVA-SciGram 7B (ours) | 80.21* | 89.93* |
| LLaVA-SciGram OneVision 7B (ours) | **83.36*** | 92.75* |

Table 5: AI2D accuracy results. Obtained by us with custom prompts: *; rest, from literature.

of our model to deeply understand the processes and elements present within a diagram. In the less demanding transparent labels split, GPT4o obtains the best result, followed by MOLMO and only 1.4 accuracy points above our best model.

### 5.2.4 ABLATION STUDY

Finally, we conduct an ablation study to better understand the impact of our dataset during training. We compare our subsets against OneVision at each phase of the pipeline in questions with image support in TQA, ScienceQA and AI2D with opaque labels. Table 6 shows how our subsets improve the results in phase 1 (alignment), 1.5 (instruction tuning) and 2 (further finetuning) in the three benchmarks with few instructions in comparison to OneVision. It also shows that training the pre-trained OneVision model with SciGram VIT and SciGram $M^3$ reports the best overall results, indicating how the combination of these subsets is required to obtain the best results.

| Phase 1 | Phase 1.5 | Phase 2 | Phase 2.5 | TQA DQ | ScienceQA IMG | AI2D Opaque |
|---|---|---|---|---|---|---|
| BLIP Alignment | - | - | - | 36.86 | 64.80 | 45.43 |
| SciGram Align | - | - | - | 50.72 | 72.68 | 59.07 |
| BLIP Alignment | Mid Stage Data | - | - | 60.49 | 74.57 | 61.66 |
| SciGram Align | SciGram VIT | - | - | 72.06 | 81.91 | 74.61 |
| BLIP Alignment | Mid Stage Data | OneVision Single-Image | - | 77.08 | 87.31 | 79.50 |
| SciGram Align | SciGram VIT | SciGram $M^3$ | - | 76.68 | 95.24 | 80.21 |
| BLIP Alignment | Mid Stage Data | OneVision Single-Image | SciGram VIT | 76.50 | 85.71 | 82.61 |
| BLIP Alignment | Mid Stage Data | OneVision Single-Image | SciGram VIT + SciGram $M^3$ | **80.21** | **97.52** | **83.36** |

Table 6: LLaVA accuracy results in datasets with diagrams.

## 6 CONCLUSION

We presented SciGram, a large-scale dataset of 200k scientific diagrams with synthetic instruction-following data, designed to improve language-vision reasoning in scientific contexts. Fine-tuning LLaVA on SciGram yields consistent gains on TQA, ScienceQA, and AI2D, outperforming or on par with the best baselines. Furthermore, incorporating SciGram into LLaVA-OneVision achieves new state-of-the-art results across all three benchmarks. Ablation studies confirm the complementary impact of alignment and instruction-tuning, with SciGram VIT and SciGram $M^3$ providing the strongest improvements. While SciGram focuses on middle-school science, the underlying methodology is general and can be applied to generate similar vision-language datasets in other domains. Limitations such as synthetic instructions and evaluation on multiple-choice benchmarks highlight opportunities for extending this work. Future research could explore richer scientific domains, integrate structured or symbolic reasoning, continue incorporating expert-verified data, and develop benchmarks that better capture multi-step diagram understanding, such as processes. By releasing SciGram and LLaVA-SciGram, we provide both a resource and a methodological foundation for advancing scientific diagram understanding.

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

# A  APPENDIX

## A.1  PROMPTS

### A.1.1  SCIENTIFIC CLAIM GENERATION

<| begin_of_text |><| start_header_id |>system<|end_header_id|>
Your task is to generate a list of 50 atomic facts that contain different combinations of the given terms.

Please follow this criteria:
− Limit yourself to the scientific domains of life sciences, earth sciences, and physical sciences at the middle school level.
− Pay attention to commonsense.
− Make the facts brief and concise yet easy to understand and meaningful.
− Use different grammatical constructions and limit length to 12 tokens or less.

Arrange your output as a jsonl file where each line is {{"claim": <fact>}}.

<| begin_of_text |><| start_header_id |>user<|end_header_id|>
TERMS: ["atom", "proton", "neutron"].

OUTPUT: {{"output": "<INSERT OUTPUT HERE>"}}

Configuration:

- Temperature: 0.7
- P value: 1.0
- Max length: 1024

### A.1.2  CAPTION GENERATION

<|im_start|>user
<| vision_start |><|image_pad|><|vision_end|> Provide a paragraph with a brief description of the diagram. Pay special attention to the main components of the diagram and the relations between them. If visible, also reflect space and temporal information, linking it to the components and relations in the diagram.<|im_end|>
<|im_start|> assistant

Configuration:

- Temperature: 0.7
- P value: 0.7
- Max length: 512

### A.1.3  MULTIPLE CHOICE QUESTION GENERATION

<|im_start|>user
<| vision_start |><|image_pad|><|vision_end|>
Formulate five multiple choice questions with 4 possible answers grounded in the diagram. The resulting questions must be middle school level questions in the subjects of life sciences, earth sciences or physical sciences. The questions should be answered using the elements from the image. For your output, follow this structure:
[{"question": <question>, "answers": {"a": <answer a>, "b": <answer b>, "c": <answer c>, "d": <answer d>}, "correct_answer": < correct_letter >}, ...].<| im_end|>
<|im_start|> assistant

Configuration:

- Temperature: 0.0
- P value: 0.0
- Max length: 1024

## A.2 LICENSING AND COPYRIGHT

As stated in our GitHub repository, related to the licensing and copyright of content linked from the SciGram dataset:

- The images linked from SciGram are copyrighted by their respective owners. The SciGram authors do not host or redistribute the images themselves.
- The URLs to the images are publicly available on the internet and not scraped from private sources.
- We respected robots.txt rules and site Terms of Service (TOS) when collecting the URLs.
- SciGram is for educational and research purposes only. The creators of SciGram do not claim ownership of the linked content.
- The SciGram dataset is licensed as Open Source, with license CC BY 4.0 `https://creativecommons.org/licenses/by/4.0`.

## A.3 DUCK DUCK GO EXAMPLES

Claim: 'Atoms are made up of protons, neutrons, and electrons.'

Query: 'Atoms are made up of protons, neutrons, and electrons diagram'

-

Claim: 'Pollution affects human health, cognitive development, and immune systems.'

Query: 'Pollution affects human health, cognitive development, and immune systems diagram'

-

Claim: 'Ocean waves can transport heat around the globe.'

Query: 'Ocean waves can transport heat around the globe diagram'

-

Claim: 'Partial melting of rocks can release minerals'

Query: 'Partial melting of rocks can release minerals diagram'

## A.4 SCIGRAM INSTRUCTION EXAMPLES

### A.4.1 SCIGRAM ALIGN

Human: I need a detailed caption for this diagram—please include explanations of each section and their functions.

Assistant: The diagram consists of two main parts: a textual explanation and a visual representation of an ionic compound. The textual explanation provides a definition of ionic compounds as giant lattice structures of ions held together by strong electrostatic forces of attraction between oppositely charged ions. It also mentions that these forces act in all directions in the lattice and are called ionic bonding. The structure of sodium chloride is used as an example to illustrate this concept. The visual representation shows a three-dimensional model of the lattice structure, with green spheres representing chloride ions and purple spheres representing sodium ions. The lattice is arranged in a cubic pattern, with ions at each corner and in the center of each face. The model also includes a key indicating the colors used to represent the ions.

-

Human: Please give a thorough caption for this diagram, covering all the important aspects and their significance.

Assistant: The diagram explains the process of how cold dark matter makes galaxies. It starts with the Big Bang, where dark matter halos (brown) collapse, tracing mass fluctuations placed during the Big Bang. As these halos collapse, gas clouds within them also collapse and begin forming

stars. These gas and young stars settle into a rotating disk, forming a proto-spiral galaxy. Stars then form in the disk, gradually building up a true spiral galaxy. The diagram also mentions numerical simulations suggesting that galaxies grow by merging halos filled with cold, non-interacting dark matter.

-

Human: Could you elaborate on this diagram with a caption that explains the concepts and relationships shown?

Assistant: The diagram shows the process of sand dune formation on a coast. It starts with an embryo dune, which is the initial stage of dune development. As the dune grows, it progresses to a fore dune, which is characterized by a gentle slope facing the sea. The yellow dune is the next stage, where the dune has developed a more pronounced shape and is further from the sea. The grey dune represents the mature stage of the dune, where it is fully formed and stable. The dune slack is the area behind the dune where vegetation can grow, providing a habitat for various plants and animals. The water table is shown as a horizontal line beneath the dune, indicating the level of groundwater. The diagram also shows the transition from the sea to the land, with the dunes gradually moving inland as they mature.

-

Human: Can you create a caption that not only labels but also explains the purpose of this diagram?

Assistant: The diagram consists of three main components arranged horizontally from left to right. On the left side, there is a cluster of small spheres, which appear to be packed closely together, forming a compact, dense structure. This cluster represents a solid state, where the spheres are fixed in place and have limited freedom of movement. Moving to the right, the second component is a beaker filled with a liquid. The spheres in the beaker are also small and closely packed, but they are suspended in a continuous medium, indicating a liquid state. The spheres are not as densely packed as in the solid state, suggesting that they have more freedom to move around within the liquid. The third component is a flask with a narrow neck, containing a column of spheres. The spheres in this flask are arranged in a vertical column, with the top sphere being at the narrow neck of the flask. This arrangement suggests a gaseous state, where the spheres are free to move and expand, occupying the entire volume of the flask. The spheres are not as densely packed as in the solid state, and they are not suspended in a continuous medium like in the liquid state. Overall, the diagram illustrates the transition from a solid state to a liquid state and then to a gaseous state, highlighting the changes in the arrangement and movement of the spheres as they move from a compact, fixed structure to a more fluid, freely moving state.

### A.4.2 SciGram VIT

Human: What is the first visible sign of the embryo's development in the diagram?

Answer choices:

a) Facial features

b) Visible elbows

c) Finger formation

d) Elongated arms and legs

Assistant: {"answer": "c"}

-

Human: What is the example of a contact force?

Answer choices:

a) Gravitational Force

b) Magnetic Force

c) Spring Force

d) Friction Force

Assistant: {"answer": "d"}

-

Human: What is the name of the part of the trap that is located at the top of the U-shape?

Answer choices:

a) inlet

b) trap weir

c) trap dip

d) trap arm

Assistant: {"answer": "a"}

-

Human: Which planet is closest to the sun?

Answer choices:

a) Mars

b) Mercury

c) Earth

d) Venus

Assistant: {"answer": "b"}

### A.4.3 SciGram VIT

Human: What is the first visible sign of the embryo's development in the diagram?

Answer choices:

a) Facial features

b) Visible elbows

c) Finger formation

d) Elongated arms and legs

Assistant: {"answer": "c"}

-

Human: What is the example of a contact force?

Answer choices:

a) Gravitational Force

b) Magnetic Force

c) Spring Force

d) Friction Force

Assistant: {"answer": "d"}

-

Human: What is the name of the part of the trap that is located at the top of the U-shape?

Answer choices:

a) inlet

b) trap weir

c) trap dip

d) trap arm

Assistant: {"answer": "a"}

-

Human: Which planet is closest to the sun?

Answer choices:

a) Mars

b) Mercury

c) Earth

d) Venus

Assistant: {"answer": "b"}

### A.4.4 SCIGRAM M³

Human: Take a look at the diagram and answer the following question by choosing one of the possible answers.

Question: "What does igneous rock become when it is subjected to heat and pressure?"

Answer choices:

a) Magma

b) Sediments

c) Metamorphic rock

d) Sedimentary rock

Arrange your output as json such as {"answer": "{<your choice>}"}. Your choice must be the associated letter to your answer.

Assistant: {"answer": "c"}

## A.5 TRAINING HYPERPARAMETERS

| Hyperparameter | Phase 1 (SciGram Align) | Phase 1.5 (SciGram VIT) | Phase 2 (SciGram M³) | Phase 2.5 (SciGram VIT) | Phase 2.5 (SciGram M³) |
|---|---|---|---|---|---|
| Epochs | 1 | 1 | 3 | 1 | 3 |
| Lora R | - | 128 | 128 | 128 | 128 |
| Lora Alpha | - | 256 | 256 | 256 | 256 |
| Vision Tower | openai/clip-vit-large-patch14-336 | openai/clip-vit-large-patch14-336 | openai/clip-vit-large-patch14-336 | google/siglip-so400m-patch14-384 | google/siglip-so400m-patch14-384 |
| mm tunable parts | mm mlp adapter | - | - | - | - |
| mm projector type | - | mlp2x gelu | | | |
| mm vision select layer | -2 | -2 | -2 | -2 | -2 |
| mm use im start end | False | False | False | False | False |
| mm use im patch token | False | False | False | False | False |
| group by modality length | - | True | True | True | True |
| image aspect ratio | - | anyres | anyres | anyres | anyres |
| mm projector lr | - | 2e-5 | 2e-5 | 2e-5 | 2e-5 |
| Image Grid Pinpoints | - | [(336, 672), (672, 336), (672, 672), (1008, 336), (336, 1008)] | [(336, 672), (672, 336), (672, 672), (1008, 336), (336, 1008)] | [(384, 768), (768, 384), (768, 768), (1152, 384), (384, 1152)] | [(384, 768), (768, 384), (768, 768), (1152, 384), (384, 1152)] |
| Batch Size | 8 | 2 | 2 | 1 | 1 |
| Gradient acc. Steps | 2 | 8 | 8 | 16 | 16 |
| Learning rate | 1e-3 | 1e-5 | 1e-5 | 1e-5 | 1e-5 |
| Weight Decay | 0. | 0. | 0. | 0. | 0. |
| Warmup Ratio | 0.03 | 0.03 | 0.03 | 0.03 | 0.03 |
| Scheduler Rate Type | cosine | cosine | cosine | cosine | cosine |
| Model Max Length | 8192 | 32768 | 32768 | 32768 | 32768 |
| Attn Implementation | sdpa | sdpa | sdpa | sdpa | sdpa |

Table 7: Hyperparameters of SciGram stages.

## A.6 SCIGRAM EXAMPLES

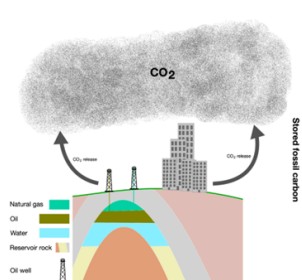
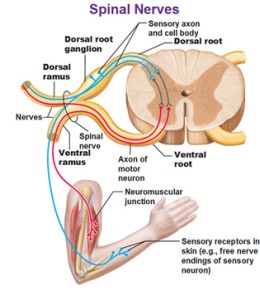
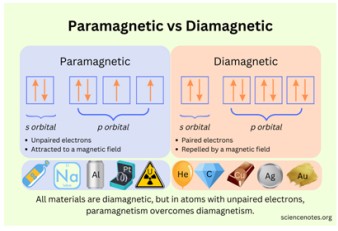

The diagram illustrates the process of carbon dioxide ($CO_2$) storage in geological formations. At the top of the image, there is a large cloud of $CO_2$, indicating the release of this greenhouse gas into the atmosphere. The $CO_2$ is then directed towards underground reservoirs, which are depicted as layers of natural gas, oil, water, and reservoir rock. The reservoir rock is shown as a large, semi-circular structure that encapsulates the other layers. \n\nOn the left side of the diagram, there is an oil well, which is connected to the reservoir rock through a pipeline. This suggests that the $CO_2$ is being injected into the reservoir through the oil well. The diagram also shows that $CO_2$ is released from the reservoir rock, indicating that the stored $CO_2$ is being extracted or vented back into the atmosphere. On the right side of the diagram, there are buildings labeled \"$CO_2$ release,\" which could represent industrial facilities or other sources of $CO_2$ emissions. The arrows connecting the $CO_2$ release from the reservoir rock to the buildings and the $CO_2$ release from the buildings to the atmosphere illustrate the cycle of $CO_2$ storage and release.

The diagram illustrates the structure and function of spinal nerves, which are crucial for transmitting signals between the central nervous system and peripheral tissues. The spinal nerve is a composite structure consisting of a dorsal root and a ventral root. The dorsal root contains sensory axons that carry information from sensory receptors in the skin and other peripheral tissues to the spinal cord. The ventral root contains axons of motor neurons that carry signals from the spinal cord to muscles and other effector organs. The spinal nerve also has a segmental arrangement, with each spinal nerve corresponding to a specific segment of the spinal cord. The diagram also shows the relationship between the spinal nerve and the spinal cord, as well as the location of sensory receptors in the skin."

The diagram titled \"Paramagnetic vs Diamagnetic\" compares the properties of paramagnetic and diamagnetic materials. On the left side, the paramagnetic category is represented with blue squares, each containing an arrow pointing up and down, symbolizing the presence of unpaired electrons in s and p orbitals. The blue squares are accompanied by a description indicating that unpaired electrons are attracted to a magnetic field. Below the paramagnetic section, various elements and compounds are shown, including oxygen, sodium, aluminum, platinum, and uranium, which are known to exhibit paramagnetic behavior due to their unpaired electrons \n\nOn the right side, the diamagnetic category is depicted with orange squares, each containing an arrow pointing up and down, representing paired electrons in s and p orbitals. The orange squares are accompanied by a description stating that paired electrons are repelled by a magnetic field. Below the diamagnetic section, helium, carbon, copper, silver, and gold are shown, which are examples of diamagnetic materials due to their paired electrons.\n\nThe diagram concludes with a statement at the bottom that all materials are diamagnetic, but in atoms with unpaired electrons, paramagnetism overcomes diamagnetism This highlights the distinction between paramagnetic and diamagnetic materials based on the presence or absence of unpaired electrons

Figure 5: Examples of SciGram Align



What is the size of the nucleus in meters?

a) 10^-10 m
b) 10^-18 m
**c) 10^-14 m**
d) 10^-15 m

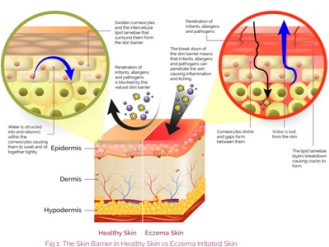

Which layer of the skin is responsible for attracting and retaining water?

a) The hypodermis and dermis together.
b) The hypodermis
**c) The epidermis**
d) The dermis

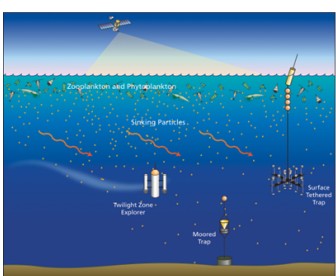

Which device is used to collect samples from the deep ocean floor?

**a) Moored Trap**
b) Surface Tethered Trap
c) Twilight Zone Explorer
d) Zooplankton & Phytoplankton

Figure 6: Examples of SciGram VIT

