# OpenReview forum: "SciGram: a large-scale dataset for scientific diagram understanding"
_ICLR.cc/2026/Conference — ICLR 2026 Conference Withdrawn Submission_

### Official Review · Reviewer_AMfe · 2025-10-26

**Soundness:** 3
**Presentation:** 3
**Contribution:** 3
**Rating:** 6
**Confidence:** 3

**Summary:**

The paper introduces SciGram, a large-scale dataset (200k+ scientific diagrams; ~1.4M multimodal instructions) aimed at scientific diagram understanding. It is built via a six-stage pipeline: terminology extraction from TQA textbooks, LLM‑generated scientific claims, web diagram retrieval, caption synthesis, MCQ synthesis, and a curated mixture with existing QA sets (TQA/AI2D/ScienceQA/ARC/OpenBookQA). Fine‑tuning LLaVA with SciGram yields strong gains and new SOTA on diagram‑centric benchmarks: TQA diagram MC, ScienceQA image‑supported subsets, and AI2D with opaque labels (Tables 3–5, pp. 8–9). Ablations show each subset contributes, with the best results from using SciGram to further tune LLaVA‑OneVision (Table 6, p. 9). Figures 5–6 (p. 23) provide concrete examples; Figure 4 (p. 7) contrasts data scale vs. OneVision.

**Strengths:**

- Targeted dataset for an underserved modality (scientific diagrams), assembled with a transparent six‑step pipeline
- Strong empirical results: SOTA or on‑par on TQA (diagram MC), ScienceQA (image‑supported), and AI2D (opaque labels) with compact domain‑focused data (Tables 3–5).
- Design choices like balanced answer letters and shuffling in curated sets to reduce superficial cues.
- Ablations and training details clarify how each subset (Align, VIT, M³) contributes (Table 6, p. 9; Table 7, p. 22).

**Weaknesses:**

- Label quality not audited: no human evaluation of caption/MCQ correctness or of whether MCQs are answerable purely from the image (Sec. 3.4–3.5).
- Potential data leakage/overlap: no deduplication analysis between web‑retrieved diagrams and evaluation sets (AI2D/ScienceQA/TQA). Shuffling answer options does not address visual content overlap.
- Licensing & reproducibility risks: images are referenced via URLs only (Appendix A.2), which can suffer link rot and variable availability; unclear long‑term reproducibility.
- Methodological gaps: heuristic thresholds (e.g., retaining images linked to ≥5 claims) lack sensitivity analysis; no error analysis by diagram type (process, topology, etc.).
- The evaluated models may need to be updated. For example, gemini 2.0 or 2.5 is necessary to be used for evaluation.

**Questions:**

- Can you provide a human audit (e.g., ~500 samples) measuring: (a) caption factuality, (b) MCQ correctness, and (c) proportion of MCQs truly answerable from the image alone?
- For baselines run with custom prompts, please share the exact prompts and any temperature/seeds; report variance across runs.
- Could you add an error analysis by diagram category (process flows, part–whole, graphs) to identify where SciGram helps most/least?

---

> ### Author Response · Authors · 2025-11-21
>
> Thank you for your positive remarks and constructive recommendations.
>
> Questions:
>
> 1. Can you provide a human audit (e.g., ~500 samples) measuring: (a) caption factuality, (b) MCQ correctness, and (c) proportion of MCQs truly answerable from the image alone?
>
> We acknowledge the lack of human evaluation on the overall quality of the dataset, and we plan to include it in the revised version. Specifically, we will assess the influence of heuristic thresholds by evaluating a set of diagram samples from our unfiltered dataset across different numbers of associated claims, as well as other filters within the dataset generation pipeline. In addition, we will evaluate the factuality , answerability, and overall quality for both SciGram Align and SciGram VIT.
>
> 2. For baselines run with custom prompts, please share the exact prompts and any temperature/seeds; report variance across runs.
>
> These hyperparameters are included in the Appendix A.1. of our paper, below each prompt. Please let us know if they are unclear, and we will clarify them further.
>
> 3. Could you add an error analysis by diagram category (process flows, part–whole, graphs) to identify where SciGram helps most/least?
>
> We agree on the importance of this point, and we will include it in the revised version.
>
> Weaknesses:
>
> 1. Label quality not audited: no human evaluation of caption/MCQ correctness or of whether MCQs are answerable purely from the image (Sec. 3.4–3.5).
>
> As mentioned in our first answer, we plan to conduct a human evaluation of our dataset quality.
>
> 2. Potential data leakage/overlap: no deduplication analysis between web retrieved diagrams and evaluation sets (AI2D/ScienceQA/TQA). Shuffling answer options does not address visual content overlap.
>
> We appreciate your comment regarding this potential bias in our dataset. We will add a deduplication stage to our pipeline to identify and remove any duplicated diagrams.
>
> 3. Licensing & reproducibility risks: images are referenced via URLs only (Appendix A.2), which can suffer link rot and variable availability; unclear long term reproducibility.
>
> We acknowledge the limitations of only providing URLs; however, this approach is necessary to publish a dataset like SciGram while complying with the licenses of the original data sources. We will explicitly discuss this limitation in the revised manuscript.
>
> 4. Methodological gaps: heuristic thresholds (e.g., retaining images linked to ≥5 claims) lack sensitivity analysis; no error analysis by diagram type (process, topology, etc.).
>
> As mentioned above, we will conduct an analysis to assess this heuristic along with other quality dimensions of our dataset.
>
> 5. The evaluated models may need to be updated. For example, gemini 2.0 or 2.5 is necessary to be used for evaluation.
>
> Due to time constraints, we were only able to evaluate the 1.5 Flash version. We plan to include other models and update to recent versions in the revised paper.

---

> > ### Comment · Reviewer_AMfe · 2025-11-22
> > **To Author Rebuttal**
> >
> > I thank the authors for their response. While some points were clarified (e.g., hyperparameters), my major concerns regarding label quality audit and data leakage remain unaddressed.
> >
> > Therefore, I am maintaining my score.

---

### Official Review · Reviewer_NmRq · 2025-10-27

**Soundness:** 3
**Presentation:** 3
**Contribution:** 2
**Rating:** 4
**Confidence:** 4

**Summary:**

The paper presents SciGram, a 200k‑diagram, ~1.37M‑instruction dataset and a six‑stage pipeline (terminology → claim generation → web retrieval → captioning → MCQ synthesis → curated mixture) targeted at scientific diagram understanding. Fine‑tuning LLaVA with SciGram yields strong gains and new SOTA on diagram‑centric portions of TQA, ScienceQA, and AI2D; ablations show each subset (Align, VIT, M³) contributes, and tuning atop OneVision performs best (Tables 3–6; Fig. 1). Training details and hyperparameters are provided; examples appear in Figs. 5–6.

**Strengths:**

- Strong empirical gains and new SOTA on diagram‑centric benchmarks (TQA diagram MC, ScienceQA with IMG support, AI2D opaque), with ablations that clarify contributions.
- Practical dataset design details (balanced answer letters; instruction formatting) that reduce superficial biases.
- Targets an underserved modality (scientific diagrams) with a transparent, reproducible six‑step pipeline; clear subset roles (Align, VIT, M³).

**Weaknesses:**

- There is a lack of human validation study of caption factuality, MCQ correctness, or image‑only answerability.
- Heuristic filters lack sensitivity analysis. Retaining only images linked to ≥5 claims may bias the dataset toward very common/templated diagrams (and possible benchmark overlap) and discard rare but educationally valuable diagrams. Without sensitivity studies, it’s unclear if results hinge on this threshold.
- Many baseline results are re‑produced with “custom prompts,” yet exact prompts, temperatures, and decoding settings are not provided, making comparisons difficult to reproduce or audit.
- The paper asserts a cost‑efficient pipeline, but does not report dollar‑costs, wall‑clock time, or GPU‑hours for data generation
- Understanding systematic errors (e.g., process diagrams vs. part‑whole, graphs with axes, occluded labels) would reveal where SciGram helps least and guide future data generation; none is provided.
- Table 6 studies subsets across stages but does not isolate: (i) the ≥5‑claims filter, (ii) number of captions/MCQs per diagram, (iii) balanced answer‑letter shuffling, or (iv) contribution of text‑only sets (ARC/OpenBookQA) within M³ to image tasks. Hence, causality of gains remains partly opaque.

**Questions:**

Please see the weaknesses above.

---

> ### Author Response · Authors · 2025-11-21
>
> Thank you for your thoughtful review and for highlighting these important aspects.
>
> Weaknesses:
>
> 1. There is a lack of human validation study of caption factuality, MCQ correctness, or image only answerability.
>
> We reckon the reviewer's concerns regarding this point. We plan to conduct studies assessing claim and caption factuality, MCQ answerability, and overall quality, incorporating humans in the loop to evaluate a representative sample of the dataset.
>
> 2. Heuristic filters lack sensitivity analysis. Retaining only images linked to ≥5 claims may bias the dataset toward very common/templated diagrams (and possible benchmark overlap) and discard rare but educationally valuable diagrams. Without sensitivity studies, it’s unclear if results hinge on this threshold.
>
> We selected the number of claims per image to align with other datasets, such as COCO, which has five captions per image. This filter also helps remove low-quality samples that are irrelevant or provide minimal information. However, we acknowledge that this rule can appear somewhat arbitrary and may introduce unintended biases. Thank you for emphasizing the need for human evaluation in this context. Taking your comments into account, we plan to evaluate a set of diagram samples from our unfiltered dataset across different numbers of associated claims. This will allow us to validate the effectiveness of the heuristic to filter out poor-quality examples or if it requires adjustment.
>
> 3. Many baseline results are re produced with “custom prompts,” yet exact prompts, temperatures, and decoding settings are not provided, making comparisons difficult to reproduce or audit.
>
> These prompts and hyperparameters are included in Appendix A.1 of our paper. We have used the same prompts for both our models and the baselines.
>
> 4. The paper asserts a cost-efficient pipeline, but does not report dollar costs, wall clock time, or GPU hours for data generation
>
> We agree that this is an important consideration and will include it in the revised version.
>
> 5. Understanding systematic errors (e.g., process diagrams vs. part whole, graphs with axes, occluded labels) would reveal where SciGram helps least and guide future data generation; none is provided.
>
> Thank you for this suggestion. We plan to provide an error analysis for the evaluated models, organized by diagram type to assess the influence of the SciGram dataset in solving different problems.
>
> 6. Table 6 studies subsets across stages but does not isolate: (i) the ≥5claims filter, (ii) number of captions/MCQs per diagram, (iii) balanced answer letter shuffling, or (iv) contribution of text only sets (ARC/OpenBookQA) within M³ to image tasks. Hence, causality of gains remains partly opaque.
>
> We appreciate your comments on the ablation study. They have been highly insightful. We will assess the impact of balanced answer-letter shuffling and the contribution of text-only MCQA datasets. Combined with a qualitative evaluation of the heuristic filters (5-claims filter and number of captions/MCQs) and the overall quality of the SciGram dataset, this study will strengthen our work.

---

### Official Review · Reviewer_J7Yw · 2025-10-31

**Soundness:** 2
**Presentation:** 2
**Contribution:** 2
**Rating:** 4
**Confidence:** 3

**Summary:**

This paper introduces **SciGram**, a new large-scale dataset specifically designed to improve **multimodal reasoning over scientific diagrams**, a long-standing challenge in vision-language research.

The authors develop a **six-stage pipeline** to construct SciGram:

1. **Terminology Extraction** – Extracts 4,820 key domain terms from middle-school science textbooks (based on the TQA dataset).
2. **Scientific Claim Generation** – Uses *LLaMA-3-8B* to synthesize 5.5M concise factual statements grounded in these domain terms.
3. **Diagram Retrieval** – Collects approximately 255K unique scientific diagrams from DuckDuckGo that align with the generated claims.
4. **Caption Synthesis** – Employs *Qwen2-VL-7B* to produce detailed textual descriptions, supporting image–text alignment.
5. **Multiple-Choice Question Generation** – Creates five reasoning-focused MCQs per diagram to assess visual comprehension.
6. **Curated Integration** – Incorporates expert datasets (TQA, ScienceQA, AI2D, ARC, OpenBookQA) to form the **M3 subset**, balancing diagram- and text-based reasoning tasks.

The final **SciGram dataset** contains approximately **1.4 million multimodal instruction samples** and over **200,000 diagrams**.

Fine-tuning **LLaVA-based architectures** on SciGram yields **state-of-the-art results** across multiple diagram reasoning benchmarks, including **TQA**, **ScienceQA**, and the **“opaque-label” split of AI2D**, demonstrating the dataset’s effectiveness in enhancing multimodal scientific understanding.

**Strengths:**

1. **Large-Scale, Domain-Specific Dataset for Scientific Diagrams**
   The paper introduces **SciGram**, the largest open-source dataset dedicated to **scientific diagram understanding**, containing over **1.4 million multimodal samples** and approximately **200,000 diagrams**.
   Unlike prior datasets (e.g., *ScienceQA*, *AI2D*, *TQA*) that combine text and natural images, SciGram focuses exclusively on **diagrammatic visual reasoning**, addressing a crucial gap in multimodal research.

2. **Transparent Data Source and Clear Licensing Claims**
   The authors provide a **well-documented public repository**, along with **explicit statements about image sources, usage rights, and permissions**.
   Such transparency is particularly important for a dataset paper, ensuring **ethical integrity**, **reproducibility**, and **legal clarity** for future research built on SciGram.

**Weaknesses:**

1. **Writing and Presentation Issues**
   -  The *Introduction* section is too brief and lacks sufficient context or motivation. Moreover, the final paragraph is redundant, as each subsequent section title already clearly indicates its content.
   -  *Figure 4* conveys very limited information but occupies a disproportionately large portion of the paper. Its inclusion could be condensed or supplemented with more meaningful analysis.
   - For a paper targeting **ICLR 2026** and discussing a relatively common task, the related work section cites only **two papers from 2025**, which is not sufficient. The survey of recent advances in multimodal and diagram reasoning is noticeably incomplete.

2. **Lack of Error and Bias Analysis**
   The paper reports overall accuracy improvements but provides little to no discussion of **failure cases**, **bias sources**, or **error typologies** (e.g., how models misinterpret symbolic diagrams or legends).
   For a dataset or benchmark paper, this type of analysis is **fundamental and expected**, as it helps the community understand where current models fail and how future work might address such weaknesses.

**Questions:**

Some general-purpose VLMs, such as QwenVL 2.5, have already achieved strong performance on ScienceQA.
Why didn’t the authors include such models for training or fine-tuning on SciGram?

---

> ### Author Response · Authors · 2025-11-21
>
> We thank you for the careful reading of our manuscript and the helpful suggestions.
>
> Questions:
>
> 1. Some general-purpose VLMs, such as QwenVL 2.5, have already achieved strong performance on ScienceQA. Why didn’t the authors include such models for training or fine-tuning on SciGram?
>
> Our current results in ScienceQA are limited to those reported in the state-of-the-art model for the overall score (T-SciQ [1]) in which QwenVL was not evaluated. QwenVL 2.5 is also absent from the official ScienceQA leaderboard, and its technical report does not provide ScienceQA results. Consequently, we do not have reliable numbers to add to the table without running our own evaluation. We would be grateful for any guidance on how to include these results mentioned by the reviewer.
>
> Weaknesses:
> 1. Writing and Presentation Issues
>
> We have taken the points you raised into consideration. In the revised version, we plan to improve the Introduction and Related Work sections to incorporate more recent and relevant work on diagram reasoning. We would also welcome any specific recommendations of papers or topics to include. In addition, as suggested, we will condense the content of Figure 4.
>
> 2. Lack of Error and Bias Analysis. The paper reports overall accuracy improvements but provides little to no discussion of failure cases, bias sources, or error typologies (e.g., how models misinterpret symbolic diagrams or legends). For a dataset or benchmark paper, this type of analysis is fundamental and expected, as it helps the community understand where current models fail and how future work might address such weaknesses.
>
> We agree that our manuscript currently lacks an error analysis, which is crucial for understanding the strengths and weaknesses of models trained with and without SciGram. We therefore plan to provide such an analysis for the evaluated models, organized by diagram type.
>
> [1] Lei Wang, Yi Hu, Jiabang He, Xing Xu, Ning Liu, Hui Liu, and Heng Tao Shen. T-sciq: teaching multimodal chain-of-thought reasoning via large language model signals for science question answering. In Proceedings of the Thirty-Eighth AAAI Conference on Artificial Intelligence and Thirty-Sixth Conference on Innovative Applications of Artificial Intelligence and Fourteenth Symposium on Educational Advances in Artificial Intelligence, AAAI’24/IAAI’24/EAAI’24. AAAI Press, 2024. ISBN 978-1-57735-887-9. doi: 10.1609/aaai.v38i17.29884.

---

### Official Review · Reviewer_Sgr8 · 2025-11-01

**Soundness:** 2
**Presentation:** 3
**Contribution:** 2
**Rating:** 2
**Confidence:** 4

**Summary:**

The research presents a scalable approach for gathering multimodal data by utilizing domain-specific scientific terminology to extract diagrams from online sources. The authors introduce SciGram, a dataset that combines visual and linguistic elements, featuring diagrams alongside synthetically created instructions within the natural sciences field. Building upon the LLaVA architecture, they develop the LLaVA-SciGram models, which represent a series of vision-language models specifically trained on the SciGram dataset.

**Strengths:**

1. The method of extracting terminology from the documents and reorganizing them into a sentence to retrieve scientific diagrams is interesting.

2. The paper is well-written and easy to follow.

**Weaknesses:**

1. Lack of scientific benchmark results. The authors should test on more recent scientific benchmarks like Mmmu[1] and Sciverse [2].

2. The paper uses Qwen2-VL-7B to generate the captions for the collected images. Therefore, it appears that the knowledge in the curated dataset originates from the scientific knowledge of Qwen2-VL-7B. So the improvement of the finetuned LLaVA may stem from the distillation of the Qwen model. Besides, this also limits the usability of the dataset. If this curated dataset is used to finetune a model more powerful than Qwen2-VL-7B, e.g., Qwen2-VL-72B or Qwen2.5-VL-7B, could it still bring improvement?

3. The paper mainly involves a dataset curation phase and an SFT phase. This paradigm has been widely explored in recent works. The novelty of this paper is limited, both in terms of data synthesis and training method.

[1] Yue, Xiang, et al. "Mmmu: A massive multi-discipline multimodal understanding and reasoning benchmark for expert agi." Proceedings of the IEEE/CVF Conference on Computer Vision and Pattern Recognition. 2024.

[2] Guo, Ziyu, et al. "Sciverse: Unveiling the knowledge comprehension and visual reasoning of lmms on multi-modal scientific problems." arXiv preprint arXiv:2503.10627 (2025).

**Questions:**

1. Can this dataset be used to improve models more powerful than Qwen2-VL-7B?

Please also refer to the weakness part.

---

> ### Author Response · Authors · 2025-11-21
>
> Thank you for your detailed review and valuable feedback.
>
> Questions:
>
> 1. Can this dataset be used to improve models more powerful than Qwen2-VL-7B?
>
> We intentionally kept the setup simple by using a single architecture and model size, as our goal was to demonstrate that the dataset itself provides measurable performance improvements. We expect these gains to generalize to other architectures and model scales, and agree that assessing this would be an interesting direction. However, given our current computational constraints, training models with 70B parameters or larger is not feasible.  For this reason, we have added to our future work the evaluation of additional architectures at the 7B scale, using SciGram to train them.
>
> Weaknesses:
>
> 1. Lack of scientific benchmark results. The authors should test on more recent scientific benchmarks like Mmmu[1] and Sciverse [2].
>
> We appreciate the reviewer’s suggestion regarding additional benchmarks. However, we believe that SciVerse and MMMU fall outside the scope of our domain focus, and therefore we do not plan to include them in our near-term roadmap. We focus on improving the model’s ability to interpret scientific textbook diagrams (e.g., cross-sections of geological layers, the cell structure, basic circuit schematics, the water cycle, etc.) at the middle-school level, which differ substantially from the image types targeted by those benchmarks.
>
> 2. The paper uses Qwen2-VL-7B to generate the captions for the collected images. Therefore, it appears that the knowledge in the curated dataset originates from the scientific knowledge of Qwen2-VL-7B. So the improvement of the finetuned LLaVA may stem from the distillation of the Qwen model. Besides, this also limits the usability of the dataset. If this curated dataset is used to finetune a model more powerful than Qwen2-VL-7B, e.g., Qwen2-VL-72B or Qwen2.5-VL-7B, could it still bring improvement?
>
> We believe that reaching state-of-the-art results on the three objective benchmarks (TQA DQ, ScienceQA IMG and AI2D Opaque), surpassing models of similar size and even API models (e.g. GPT4o or Gemini), supports the effectiveness of the SciGram dataset in improving model performance within the target domain. Furthermore, our dataset-construction methodology is sufficiently generalizable to be applied to other architectures. As noted in our earlier response, we also plan to expand our set of models trained with SciGram to include additional architectures.
>
> 3. The paper mainly involves a dataset curation phase and an SFT phase. This paradigm has been widely explored in recent works. The novelty of this paper is limited, both in terms of data synthesis and training method.
>
> The novelty of our approach lies in the dataset-generation methodology itself, which enables the creation of a large synthetic dataset using only the core vocabulary of the domain.

---

### Note · Authors · 2026-03-04

I have read and agree with the venue's withdrawal policy on behalf of myself and my co-authors.

---

### Meta-Review · Area_Chair_ZxYM · 2026-01-07

**Summary:**

Paper was reviewed by for reviewers and received: 2 x marginally below the acceptance threshold, 1 x reject and 1 x marginally above the acceptance threshold ratings. The overall sentiment of reviews was marginally negative to strongly negative. The main concerns outlined by reviewers focused on:

(1) Lack of result reporting on standard scientific benchmarks, e.g., Mmmu and Sciverse [Sgr8]
(2) Potential confounding (distillation) effects from data generation pipeline that leverages Qwen2-VL-7B [Sgr8]
(3) Limited novelty focusing on data curation for SFT [Sgr8]
(4) Concerns with exposition and presentation [J7Yw, AMfe]
(5) Lack of error and bias analysis [J7Yw]
(6) Lack of human validation of the dataset [NmRq, AMfe]
(7) Evaluation is lacking rigor (with missing prompts, lacking cost analysis, etc.) [NmRq]
(8) Potential issues with data leakage / overlap [AMfe]
(9) Potential concerns with reproducibility [AMfe]

Authors have largely acknowledged these concerns in the rebuttal, in many cases promising to make changes in the paper revision (without yet doing so). As a result, AC sees no other recourse as to recommend Rejecting the paper at this time, giving an opportunity to authors to implement required changes to strengthen the work.

**Reviewer Concerns:**

(1) Lack of result reporting on standard scientific benchmarks, e.g., Mmmu and Sciverse [Sgr8]
(2) Potential confounding (distillation) effects from data generation pipeline that leverages Qwen2-VL-7B [Sgr8]
(3) Limited novelty focusing on data curation for SFT [Sgr8]
(4) Concerns with exposition and presentation [J7Yw, AMfe]
(5) Lack of error and bias analysis [J7Yw]
(6) Lack of human validation of the dataset [NmRq, AMfe]
(7) Evaluation is lacking rigor (with missing prompts, lacking cost analysis, etc.) [NmRq]
(8) Potential issues with data leakage / overlap [AMfe]
(9) Potential concerns with reproducibility [AMfe]

**Reviewer Scores:**

[AMfe] explicitly notes that rebuttal does not do enough to warrant upgrading his/her score in discussion. AC does not believe other reviewers would make changes to their scores either given the relatively sparse nature of the rebuttal.

---

### Decision · Program_Chairs · 2026-01-26

Reject